# Telemedicine Management: Approaches and Perspectives—A Review of the Scientific Literature of the Last 10 Years

**DOI:** 10.3390/bs13030255

**Published:** 2023-03-14

**Authors:** Joaquín Aguirre-Sosa, Jorge Alberto Vargas-Merino

**Affiliations:** 1Faculty of Business, School of Management, Universidad Privada del Norte, San Juan de Lurigancho, Lima 15434, Peru; n00220615@upn.pe; 2Faculty of Business, Department of Research, Innovation and Social Responsibility, Universidad Privada del Norte, San Juan de Lurigancho, Lima 15434, Peru

**Keywords:** telemedicine, eHealth, telehealth, perspectives

## Abstract

This research paper describes the approaches and perspectives of telemedicine worldwide. The objective of this literature review was to analyze the theoretical and empirical studies that address telemedicine management in the last 10 years based on the scientific literature obtained from the Scopus, Scielo, Ebsco, ProQuest, Dialnet, and Redalyc databases, from which 50 articles were selected. The inclusion criteria were the last 10 years, scientific articles, language, variables, and open access. The non-inclusion criteria were repeated articles, not addressing the variable, and not open access. The results show a tendency to manage telemedicine through various approaches and scenarios. These can be grouped into humanistic, socioeconomic, ethical, contingency in the Armed Forces—NASA, and application in the field of medicine with teaching to the entire chain of users, as well as controls and monitoring of patients. In this sense, it is concluded that telemedicine management worldwide faces challenges that must be overcome to reduce still-existing barriers and achieve better access to health systems worldwide.

## 1. Introduction

Currently, telemedicine has a social context meaning. It is the technology adapted to medicine, health, patients, organizations, and the country. It is the use of information and communication tools in the service of health. Likewise, telemedicine is a process, not a technology, and it has evolved hand in hand with the development of technology, involving providers, users, and health organizations [1].

The concept of telemedicine worldwide has gone through multiple existing definitions that have evolved. Etymologically, the term telemedicine comes from the Greek word ‘tele,’ which means at a distance, and from the Greek word ‘mederi’ or from the Latin ‘medecus,’ which means to cure. From this, it can be inferred that the nature of telemedicine is the management of services with information to users in their own environment, which requires new tools to overcome socioeconomic, cultural, and geographical barriers [2].

Telemedicine, on the other hand, is being used much more frequently worldwide. It has achieved a very popular position in North America, Australia, South Africa, and the Scandinavian countries; however, in Hungary, it has not been established, despite many attempts, so experts had to analyze the ethical, legal, and economic aspects of telemedicine from the point of view of physicians and patients. The conclusion was that it is not clear whether telemedicine is worthwhile to apply in Hungary at present because of the lack of experience gained in the Hungarian environment [3].

Telemedicine is viewed by the scientific community as a sustainable and high-quality means of ensuring the welfare of populations with long life expectancies. In addition, it brings medical care to remote and hard-to-reach places in poor and emerging countries. However, there are still challenges and barriers to overcome, such as resistance to the adoption of the innovative model of telemedicine, its applications, the lack of studies reporting economic benefits, and the absence of an international legal framework that facilitates professionals’ ability to provide services in different jurisdictions and countries [4].

The space race contributed a lot to the development of telemedicine. The National Aeronautics and Space Administration (NASA) implemented telemetry and telemonitoring systems for its astronauts in 1958 [5]. Likewise, in the Spanish Armed Forces, telemedicine, following the operational model of military healthcare, managed to achieve sustained development in a few years [4]. Furthermore, in Latin America, multi-sector projects requiring great efforts to implement and apply telemedicine are carried out due to the geographic and socioeconomic conditions of the populations [6].

In this context, the following question arises: What is known about telemedicine management in the last 10 years? Furthermore, how has its implementation been managed worldwide, and what expectations and approaches have been set? Therefore, this literature review aims to analyze theoretical and empirical studies addressing telemedicine management in the last 10 years. This research will define the approaches and perspectives so that this study will serve as a common thread for future research.

Likewise, conducting this research is justified because, at present, telemedicine was catapulted with the arrival of SARS-CoV-2, reaching development and a mandatory and impressive implementation in all organizations worldwide because, as the pandemic elapses, the interest in and use of telemedicine increases [1]. Telemedicine management worldwide has also generated many expectations and has focused on five main approaches: humanistic, socioeconomic, ethical, contingency, and in the field of medicine. 

The importance of carrying out a literature review of telemedicine is currently transcendental because with SARS-CoV-2, it was implemented abruptly in all health systems in the world, and little by little it was improved and implemented to attend to the health emergency presented. In this sense, the COVID-19 pandemic forced the promotion of the use of telemedicine in different areas of health, which, thanks to current technology, provided new services, innovating and complementing emergency and ambulatory care for the emergency the world is going through [7]. Moreover, telemedicine is here to stay and therefore must be implemented using many approaches in order to establish protocols that make it a global tool for all citizens of the world in whatever remote region they are in. Therefore, it requires attention from different perspectives in order to fill existing gaps such as ethical and legal ones, which will guarantee the implementation of a telemedicine service in the world. 

The document is structured as follows. Section 2 presents the methodological criteria. Section 3 covers both descriptive and narrative results, showing relevant graphs of the systematized articles and their views linked to the research questions. Finally, Section 4 develops the discussion and conclusions, presenting a critical analysis of the findings.

## 2. Materials and Methods

This research is a literature review, which, according to [8], is a process of compiling available information into clear and structured summaries that aim to answer a specific question. Consisting of multiple articles and sources of information, they represent the highest level of the hierarchy of evidence. Thus, this review is important because it identifies what is known about the topic, what has been researched, what are the most salient developments in a given period, and what aspects remain unknown in a way that allows the research question to be answered.

Literature reviews are important in many critical and useful aspects of the evolution of the world’s systems. They provide a synthesis of the state of knowledge in a given area, from which important and future research priorities can be identified [9,10].

This review began with a search in each of the databases, with their respective keywords and search strings or strategies of interest for the large variable and/or category of study. Initially, a large amount of research was found, so a first selection was made where the search strategies were expressly contained in the title, leaving a smaller number, together with several inclusion and exclusion criteria, such as the years of publication, the content of the categories to be analyzed, and the availability of each article (open access), among others. 

Throughout this process, Microsoft Excel was used to organize all the downloads allowed by the databases and their respective abstracts; with the help of text formulas and internal searches, the articles were pre-selected for reading and downloading according to DOI or URL link, respecting the criteria and objectives of the research, and also excluding duplicate articles. This was followed by a more exhaustive review, i.e., articles were selected by approach and, within each approach, the content of each article was prioritized. Finally, we worked with 50 articles addressing telemedicine from 5 approaches over the last 10 years. These databases were Scopus, Ebsco, Scielo, ProQuest, Redalyc, and Dialnet; the time period covered the years 2012–2022.

In the following, we show the search terms used (as a grouped report of everything searched), highlighting their relevance to the research question and objective, and discovering their different approaches and perspectives in the different databases: telemedicine, telemedicine management, telehealth, eHealth, implementation, telemedicine system. all of which can be seen in Figure 1.

## 3. Results

After searching the information in the databases, 50 articles were obtained as a final result valid for the literature review. These were ordered in a classification matrix according to authors, year of publication, country of origin, and database (see Table 1). 

Table 1 shows the authors of the 50 articles selected for the analysis of the results. In addition, the year of publication, the country of origin, and the database where the article was selected are shown.

Figure 2 is a world map showing the geographical location of the studies addressed. This has made it possible to identify the approaches and perspectives across the articles chosen for this literature review.

In the literature reviewed, it has been possible to identify the approaches and perspectives managed in telemedicine worldwide during the last 10 years. There are five approaches: ethical, humanistic, socioeconomic, contingency in the Armed Forces and NASA, and the application in the entire chain of users in the field of medicine, from monitoring and control of various pathologies to teaching and testing. 

### 3.1. Approaches

#### 3.1.1. Ethical Approach

Ethics in telemedicine has been one of the most controversial factors due to the physician’s responsibility towards the patient in different spaces and to provide a legal response to the practice of telemedicine worldwide. In this way, risks to the quality, safety, and continuity of medical care are avoided due to the implications of the intervention on a person’s health. Therefore, it is very important for professionals who use telemedicine to know ethics and law due to the implications in terms of criminal liability in what they do. A statement on ethical issues will always be encouraged [21,34,48,51,53].

#### 3.1.2. Humanistic Approach

Telemedicine acquires its primary purpose of creation: to favor equity by promoting accessibility for all, regardless of their place of residence, resources, or reality, because telemedicine is a set of valid strategies to improve access to health services for all citizens worldwide. In this sense, telemedicine should be a tool that facilitates healthcare, offering all its advantages to these professionals without being tediously or excessively complex. It should arise from the need to improve the health of the population through the demand for doctors in all corners of the world who are prepared and trained to provide greater rigor in the diagnosis and treatment of their patients. All the experiences described in the social approach to telemedicine in the last 10 years call on all political leaders and key players in the states to work collaboratively, with institutional efforts to cover all levels of society, with a systemic approach that enhances what has already been developed to date. In this way, the relevant findings found in Africa, Latin America, and other places should be raised to overcome the existing gaps, with the firm purpose of implementing telemedicine in all countries around the world. In this sense, social factors are perceived as the most complex. The implementation of telemedicine has been delayed for several reasons, including repeated and persistent resistance from stakeholders, legal constraints, patient indifference, the lack of a consolidated rationale, and disharmony [13,14,22,23,25,26,29,30,33,37,45,52,54]. 

#### 3.1.3. Contingency Approach

Since its origins, telemedicine has been managed within the Armed Forces all over the world as contingencies to face real scenarios of support in combat fields or rescues from inhospitable places. The National Aeronautics and Space Administration (NASA) has also implemented it since its origins to monitor the health of astronauts. The contributions of this approach are presented below (Table 2).

Table 2 shows the most relevant studies that address telemedicine from a contingency approach based on the experience of the military in the Armed Forces and the experience of NASA.

#### 3.1.4. Socioeconomic Approach

Currently, in many countries around the world, the socioeconomic situation of their populations is such that many are distributed throughout territories with difficult access, in many cases due to geographical conditions; this is coupled with existing limitations regarding access to medical care, such as scarcity of resources, and the physical and cultural distance between the public supply and the demanding population. In addition, experiences in countries such as Bangladesh, which has made significant improvements in its healthcare system; however, the country still faces public health challenges, with limited and unequal access to healthcare services and insufficient resources to meet the demands of the population. Likewise, telemedicine has not been routinely used in Hungary despite several attempts. This is presumably due to the government’s previous dismissive attitude to this methodology. The development of telemedicine in Chile is not so different from that observed in the United States. It presents the same barriers: accessibility, cybersecurity, and medical liability. There is also no uniform and specific regulation. In this sense of ideas, telemedicine in a public health system requires the definition of a clear and stable business model which incorporates telemedicine in the service portfolio of the administration and offers health organizations the possibility of obtaining reimbursement for the activity performed. In addition, it can contribute to improving comprehensive health care for the population, but it has to overcome the problems of adherence to the intervention, particularly concerning costs. Furthermore, it should be clear that the benefits of telemedicine development should be reflected in health systems worldwide, which should understand that telemedicine should be an innovative strategy for effective change management [3,11,12,17,27,35,36,39,56,57]. 

#### 3.1.5. Medical Approach: Experiences of Interventions around the World

Telemedicine is being managed by leaps and bounds worldwide throughout the entire healthcare user chain, from monitoring, control, intervention, or consultation. It depends on the pathology addressed with the technological tools available to health professionals (see Table 3). 

Table 3 shows all the authors addressed in the approach to care throughout the health service chain, to monitor and control various pathologies and also to teach.

Currently, the use of information and communication technologies is a major challenge in health systems worldwide. With the coronavirus pandemic, regions around the world have implemented telemedicine services, with varying degrees of success, underlining the need for infrastructure, investment, and regulation in this regard.

Telemedicine is about providing health services, where distance is an important component, by any health professional, using new communication technologies for the valid exchange of information in diagnosis, procedure, and prevention of pathologies or injuries, research and evaluation, and continuing education of health providers, all in the interest of improving the health of individuals and communities.

In terms of human factors, the main challenge facing telemedicine is overcoming resistance to change, which is often multifactorial in nature, such as lack of training in the use of information technologies, fear of the unknown, and ethical and legal issues. It is also a reorganization of health systems due to the fact that most health professionals are used to providing face-to-face care, where there is not much experience in distance care, due to lack of education and training in most cases. Now it is also about the workload that health professionals struggle with and their particular interests and beliefs, i.e., what they advocate and put into practice, versus those who do not believe in these approaches. Moreover, these new tasks require training.

Their proper use requires a plan that involves the entire health system, as well as public policies, adequate legislation, and health institutions with the appropriate infrastructure to ensure connectivity and relevant communication sets. All of this involves time and financial resources in addition to the above components.

The adoption of telemedicine in all health systems around the world, following the global SARS-CoV-2 crisis, demonstrates its valuable utility as an effective tool to safeguard the so-called social distancing in clinical settings. However, this situation is evidence of the slow adoption of telemedicine throughout history, despite the varied research that has been carried out, as a result of the experience acquired in its application and adoption in different healthcare scenarios, ranging from monitoring to teaching. For this reason, the usefulness of telemedicine should be valued, and it should not only be limited to the management of the current health crisis but should also be transcended because it has been established abruptly, adapted to the crisis, and is here to stay. In this sense, a large number of outpatient consultations in various settings can be managed clinically and efficiently through telemedicine. It also requires the necessary infrastructure with the use of technology, and the necessary logistics in terms of material, human, and financial resources from the point of view of governments for public health centers and also private investment.

Today, due to the rise of telemedicine, the main actors who have worked in this field cannot rest on their laurels; it is necessary to develop a know-how, with all the events they have experienced, and which allowed them the merit of having a vision that enabled them to extend the reach of healthcare resources to those who need them, regardless of distance and time. It is time to maintain the route to safe and effective medical care. More specifically, it is time to use telemedicine outreach to all, for all actors and especially physicians, who must respect standards, institutional protocols, quality assurance mechanisms in place, prompt reporting of adverse events, proper documentation, and follow-up through virtual health records [59].

During the height of the global health crisis, it was reported that some consultations could be a vector for SARS-CoV-2 transmission. Therefore, face-to-face interventions had to be postponed during the pandemic period. In this context, the importance of the adoption, implementation, and application of telemedicine in consultations arises. This know-how can help all health systems to ensure telemedicine with global reach that is increasingly committed to meeting the needs of populations in all settings around the world [60].

## 4. Discussion

The approaches and perspectives of telemedicine management worldwide in the last 10 years are conceptualized in each of the articles analyzed, identifying five approaches: ethical, humanistic, socioeconomic, contingency, and support to the whole medical system. The literature review is composed of articles published between 2012 to 2022, and 50 empirical articles have been selected, analyzed, and synthesized. In this sense, in any intervention in the health of a person, not only is the technical possibility at stake, but it must also comply with acceptable professional criteria framed in ethics and law [51]. In addition, political cooperation and cooperation of the entire health sector should be sought to take action and implement telemedicine with innovative care models, based on evidence of its effectiveness, to ensure access to health for all populations in the world [14]. 

The exploration of all 50 articles of scientific literature places us in a context of a diversity of approaches and perspectives that have originated throughout the experience in the world and also hand in hand with the growth of technology, which is the main bastion for the telemedicine—technology pairing to advance and achieve their purpose of creation. In this sense, ethics in telemedicine has been one of the most controversial factors. The responsibility regarding the patient lies on those who are present in the same place as the patient, providing health care. It is different for someone physically present with the patient, using telemedicine as a tool to obtain a specialized concept to cure a disease [21]. 

On the other hand, we have the socioeconomic approach, which is supported by [61], who state that the impact of the COVID-19 pandemic aggravates the vulnerability of rural areas. Access to health care is an important variable in health for the health organization that must provide timely emergency services; health care is an urgent and socially relevant issue in all countries. Therefore, the use of telemedicine in rural health care in times of pandemics for disease control is one of the major issues facing rural communities due to the scarcity of resources, the long distances over which they are geographically located, and the deterioration of health care facilities evidenced in recent years. In this sense, telemedicine is called upon to solve this problem and, at the same time, to help reduce the health gaps that exist between rural and urban areas, which is the great disparity between access to health care and general health of rural inhabitants compared to urban dwellers.

According to the classification of approaches, the contingency approach is based on the experience of military strategies to face challenges on internal and external fronts and the vision they had to implement it a long time ago, which places them as pioneers in the use of telemedicine. Likewise, in the United States, one of the first uses of telemedicine was established by the National Aeronautics and Space Administration (NASA) in 1960 to monitor astronauts in flight by doctors and medical teams during their mission [16,39,46,51].

On the other hand, Mohammadi et al. show the application of telemedicine in various remote locations by the Armed Forces. In other words, military medicine is an academic discipline with broad applications and fields. Military physicians often practice a specific body of knowledge of the medical problems and needs of the Armed Forces, which is often different from general medicine. Furthermore, in their results, the researchers found that videoconferencing and e-mail were the main means of communication in telemedicine. Therefore, military medicine, given the shortage of specialists and human resources, needs technology and should be equipped with videoconferencing equipment to improve the quality of health services. Further, the lack of studies related to telemedicine in military medicine was one of the main limitations of this research [62].

Telemedicine can be helpful in several areas: acquisition, storage, concentration, and dissemination of knowledge; provision of health services; scientific research applied to diseases and health technologies; and data collection and management. In this sense, the socioeconomic approach is based on the assertion that to implement telemedicine, legislation is needed to address the issue of the possible personal responsibilities of those who act in the sometimes confusing context of these consultations, often amid more uncertainty than in face-to-face care. A regulation specifying the requirements for this type of medical care is also required. In addition, the definitive incorporation of telemedicine into routine clinical practice in a public health system based on the provision of services needs the definition of a clear and stable business model that incorporates telemedicine into the administration’s portfolio of services and reduces inequalities in access to health, in a service based on new technology, and an understanding that telemedicine must be an innovative strategy for effective change management [3,12,17,24,27,35,49,53,56]. 

The humanistic approach is based on the fundamental rights of human beings, and when humanism in health is mentioned, it refers to the attitude of treating people with dignity and respect, to models of relationship and assistance focused on the person, and to the treatment of the human being as a whole, in a comprehensive manner [2,13,14,22,23,25,26,29,37,43,45,52,54]. 

Furthermore, this approach is supported by a study by [63] who argue that pandemics pose significant challenges to health care, especially in vulnerable countries such as those in Latin America, which experienced during SARS-CoV-2 a high occupational risk generated by the saturation of health services and the shortage of personal protective equipment (PPE) for health care workers. This required the implementation of strategies to respond efficiently to the health emergency. Compared to developed countries, which had more experience, telemedicine was practiced in parallel to conventional care long before the pandemic. However, in developing countries, the absence of technological resources and lack of platforms for telemedicine consultations hindered its use and development.

Another approach with high expectations in telemedicine corresponds to the field of action of the medicine. All that is known is broad, with thousands of applications in all pathologies and the technology available in each country. In this sense, in one experience, multicomponent face-to-face diabetes care services were successfully transformed into the telemedicine modality. At the primary healthcare level, many patients with type 2 diabetes mellitus may be candidates for being included in this and other types of telemedicine strategies in health contingency scenarios. Likewise, knowing how patients feel about telemedicine can help to better define the use of this type of care. Most patients express a high degree of satisfaction with the health care received and with the possibilities of contact and accompaniment, emphasizing the effort made by the health care professionals to continue monitoring oncology patients. There are many experiences accumulated in the various fields of health care. These experiences will continue to be enriched, and their methodologies of approaching patients will be improved to achieve development in the whole health system [15,18,20,32,40]. 

On the other hand, the medical approach is the one that has the most development in the world through different telemedicine interventions globally, thus [64] in a study sought to determine the effectiveness of telemedicine in the provision of diabetes healthcare services in low- and lower-middle-income countries. They conducted a selective literature search, extracting data on study characteristics, key endpoints, and outcome effect sizes. Then, using random effects analysis, they performed a series of meta-analyses for both biochemical outcomes and patient-related properties. They concluded that although telemedicine was found to be effective in improving several diabetes-related outcomes, the certainty of the evidence was very low due to considerable heterogeneity and risk of bias.

This study is a theoretical contribution to clarify which implications of telemedicine should be considered in order to create a frame of reference to understand what it means to manage its approaches and perspectives where there are areas with a remarkable development in its use, while in others it is still slow and fragmented. It is hoped that in this modality, its practical application will be particularly enhanced. This request arises from the need to create a common framework that can be adapted to different settings and provides clear guidance on how to understand telemedicine management. At the same time, it provides analytical perspectives related to the multiple approaches and viewpoints that frame it, highlighting the overall impact of coronavirus on the process of use and acclimatization to telemedicine, the components that make it feasible, and the barriers that impede it. 

It is worth mentioning the main limitation of this file. We start with the fact that we are talking about a literature review that, despite searching as many parts of the literature as possible, could not be fully entered into several databases such as Wiley. In addition, by not including conference papers, field reports, organizational reports, etc., we have not been able to include any of the associated literature. Although we strive to minimize author subjectivity, it is possible to assume that important references have been ignored for research purposes when selecting journals with high scientific impact. Despite its limitations, it is hoped that this research will provide important insights for the formulation of further experimental and empirical research.

## 5. Conclusions

The approaches and perspectives of telemedicine management worldwide in the last 10 years are conceptualized in each of the articles analyzed, identifying five approaches: ethical, humanistic, socioeconomic, contingency, and support to the whole medical system. The literature review is composed of articles published between 2012 to 2022, and 50 empirical articles have been selected, analyzed, and synthesized. In this sense, in any intervention in the health of a person, not only is the technical possibility at stake, but it must also comply with acceptable professional criteria framed in ethics and law [53]. In addition, political cooperation and cooperation of the entire health sector should be sought to take action and implement telemedicine with innovative care models, based on evidence of its effectiveness, to ensure access to health for all populations in the world [14]. 

The exploration of all 50 articles of scientific literature places us in a context of a diversity of approaches and perspectives that have originated throughout the experience in the world and also hand in hand with the growth of technology that is the main bastion for telemedicine—technology pairing to advance and achieve their purpose of creation. In this sense, ethics in telemedicine has been one of the most controversial factors. The responsibility regarding the patient lies on those who are present in the same place as the patient, providing health care. It is different for someone physically present with the patient, using telemedicine as a tool to obtain a specialized concept to cure a disease [21]. 

Telemedicine can be helpful in several areas: acquisition, storage, concentration, and dissemination of knowledge; provision of health services; scientific research applied to diseases and health technologies; and data collection and management. In this sense, the socioeconomic approach is based on the assertion that to implement telemedicine, legislation is needed to address the issue of the possible personal responsibilities of those who act in the sometimes confusing context of these consultations, often amid more uncertainty than in face-to-face care. A regulation specifying the requirements for this type of medical care is also required. In addition, the definitive incorporation of telemedicine into routine clinical practice in a public health system based on the provision of services needs the definition of a clear and stable business model that incorporates telemedicine into the administration’s portfolio of services and reduces inequalities in access to health, in a service based on new technology, and an understanding that telemedicine must be an innovative strategy for effective change management [3,12,17,24,27,35,49,53,56].

Another approach with high expectations in telemedicine corresponds to the field of action of the medicine. All that is known is broad, with thousands of applications in all pathologies and the technology available in each country. In this sense, in one experience, multicomponent face-to-face diabetes care services were successfully transformed into the telemedicine modality. At the primary healthcare level, many patients with type 2 diabetes mellitus may be candidates for being included in this and other types of telemedicine strategies in health contingency scenarios. Likewise, knowing how patients feel about telemedicine can help to better define the use of this type of care. Most patients express a high degree of satisfaction with the health care received and with the possibilities of contact and accompaniment, emphasizing the effort made by health care professionals to continue monitoring oncology patients. There are many experiences accumulated in the various fields of health care. These experiences will continue to be enriched, and their methodologies of approaching patients will be improved to achieve development in the whole health system [15,18,20,32,40]. 

## Figures and Tables

**Figure 1 behavsci-13-00255-f001:**
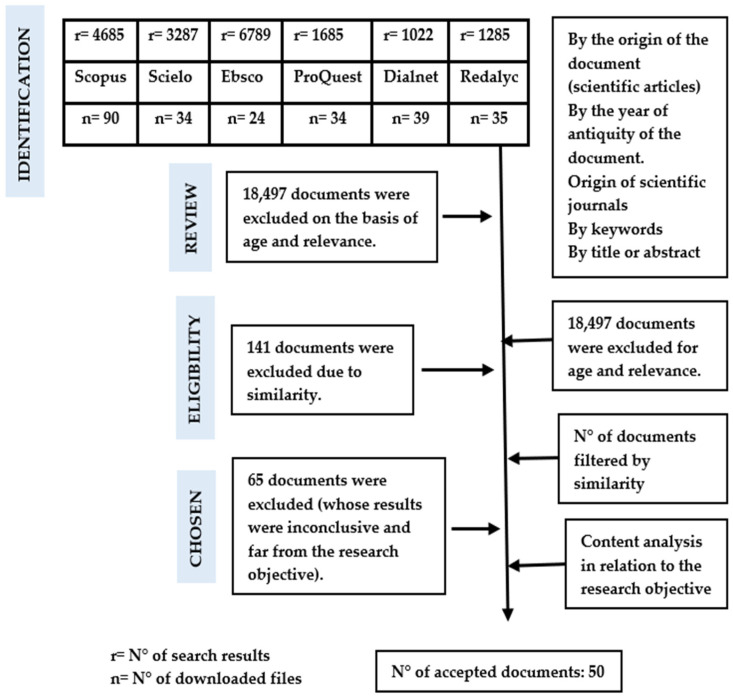
Graphical representation of the review process.

**Figure 2 behavsci-13-00255-f002:**
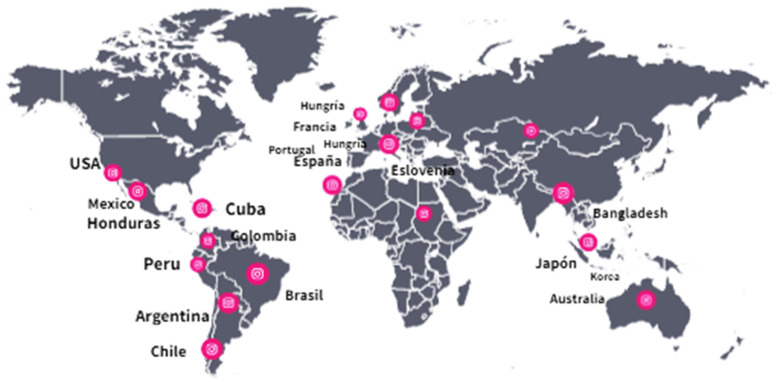
Geographical location of the articles reviewed.

**Table 1 behavsci-13-00255-t001:** General information on the articles found.

Authors	Year	Country	Database
[11]	2012	France	Scielo
[12]	20212	Spain	Scielo
[13]	2012	Spain	Scopus
[14]	2013	Belgium	Scopus
[15]	2013	Colombia	ProQuest
[16]	2013	USA	Scopus
[3]	2013	Hungary	Scopus
[17]	2013	Argentina	Scielo
[18]	2014	Egypt	Scopus
[19]	2014	Spain	Scopus
[20]	2014	Spain	Scielo
[21]	2014	Mexico	Ebsco
[22]	2014	Spain	Scopus
[23]	2015	Peru	Ebsco
[24]	2015	USA	Ebsco
[25]	2015	Spain	Scopus
[26]	2015	Colombia	ProQuest
[27]	2015	Bangladesh	Scielo
[28]	2015	Colombia	Scopus
[29]	2016	Brazil	Scielo
[30]	2016	Spain	Dialnet
[31]	2016	Brazil	Dialnet
[32]	2016	Colombia	ProQuest
[33]	2017	Colombia	ProQuest
[34]	2017	Colombia	ProQuest
[35]	2017	USA	Ebsco
[36]	2017	Portugal	Scopus
[37]	2018	Australia	Scopus
[38]	2018	Australia	ProQuest
[39]	2018	Spain	Scielo
[40]	2018	Cuba	Redalyc
[41]	2019	Slovenia	Scielo
[42]	2019	Spain	Scielo
[43]	2019	Brazil	Scielo
[44]	2020	Venezuela	Redalyc
[45]	2020	Honduras	Redalyc
[46]	2020	USA	Scopus
[47]	2020	Spain	Scielo
[48]	2021	Mexico	Scielo
[49]	2021	Chile	Scopus
[50]	2021	Mexico	Scopus
[51]	2021	Spain	Scopus
[52]	2021	Spain	Scopus
[53]	2021	Chile	Scopus
[54]	2021	Korea	Redalyc
[2]	2021	Argentina	Ebsco
[55]	2021	Japan	Scopus
[56]	2021	Mexico	ProQuest
[57]	2021	Spain	ProQuest
[58]	2022	Spain	Scopus

**Table 2 behavsci-13-00255-t002:** Contingency approach in the Armed Forces and NASA.

Author (s)	Origin and Year	Contributions
[16]	U.S.A 2013	The U.S. Army Medical Department’s telemedicine network spans 50 countries and territories, from American Samoa to Afghanistan, in 19 time zones. In total, 22 service lines are available, of which behavioral health makes up 55% of all telemedicine services.
[39]	Spain 2018	The experimentation of teleconsultation from the base camp of Gasherbrum II (Pakistani Himalayas) to the Military Hospital “Gomez Ulla” allowed the development of a telemedicine kit that will simplify the inconveniences for interventions in hostile camps.
[43]	Spain 2019	Studies of international missions in which the Spanish Armed Forces and their allies have implemented a facility with Role 1 health capabilities to evaluate telemedicine as a key tool to support doctors deployed in missions.
[46]	U.S.A 2020	In the United States, one of the first uses of telemedicine was established by the National Aeronautics and Space Administration (NASA) in 1960 to monitor astronauts in flight by physicians and medical teams during their Project Mercury mission.

**Table 3 behavsci-13-00255-t003:** Contributions of the application approach to the medical field throughout the user chain.

Author (s)	Origin and Year	Contributions
[19]	Spain 2014	In a telemedicine program, 410 patients were intervened by telephone. It was reported that for 288 patients (89.4%), the devices were easy to operate at home by themselves. Only in 12 cases (3.7%) did the patient consider the telemedicine device to be a difficult workload to integrate into their daily life.
[18]	Egypt 2014	This work proposes the design and implementation of a wireless telemedicine system in which all physiological vital signs are transmitted to a remote medical server via mobile networks, in case of emergency, and the Internet, in the case of long-term monitoring.
[15]	Colombia 2013	Positive perception by the medical team regarding the visual environment of the system, its functionality, and relevance for post-operative care in ambulatory surgery patients.
[20]	Spain 2014	Parents or legal guardians in pediatrics use new technologies and find it convenient to be able to ask about health issues that do not require a face-to-face assessment with the professionals who regularly attend to their child without having to make an appointment and travel to the health center.
[28]	Colombia 2015	Importance of evaluating the different models of telemedicine assistance for the maternal–fetal pairing to identify the lessons learned and the success factors necessary to implement new models.
[31]	Colombia 2016	Through scientific and technological cooperation between Brazil and Colombia for practicing telemedicine. Currently, undergraduate and graduate students and professionals in the health field are integrated for developing models, strategies, and mechanisms to transmit information that takes into account distance and digital safety.
[32]	Colombia 2016	Telemedicine and telehealth are not only an alternative for dealing with complex chronic pathologies such as heart failure; they have become a necessity in our environment.
[38]	Australia 2018	Digital interventions can be used by clinicians as an adjunct service during post-treatment care for melanoma, especially in regions with fewer resources of healthcare professionals and infrastructure, such as rural and remote Australia.
[41]	Slovenia 2019	A case of good practice that demonstrates that in Slovenia, as elsewhere, the combination of appropriate technology, qualified staff, good management, and sufficient funding can result in telemedicine delivering healthcare services more optimally than the traditional way.
[44]	Venezuela 2020	Psychosocial and behavioral improvements in self-management efficacy have been demonstrated in young patients with type 1 diabetes who participate in telemedicine compared to those who attend face-to-face visits.
[42]	Spain 2019	The use of clinical pain protocols in clinical practice is a good example of information processing in the healthcare field by applying the benefits of telemedicine.
[47]	Spain 2020	TELEA appears to be a good tool for monitoring patients with COVID-19 in home confinement. Nursing care is essential to the success of telecare with TELEA.
[55]	Japan 2022	The design and implementation of a wireless telemedicine system are proposed in Japan for remote monitoring.
[50]	Mexico 2021	A redesign of the care model with the incorporation of technologies and telemedicine should be considered to mitigate chronic disease morbidity and mortality during the COVID-19 pandemic and in the post-COVID-19 era.
[58]	Spain 2022	Most patients express a high degree of satisfaction with the health care received and with the possibilities of contact and accompaniment, emphasizing the effort made by health care providers to continue monitoring oncology patients.

## Data Availability

Not applicable.

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
