# Peer review of "Telemedicine Management: Approaches and Perspectives—A Review of the Scientific Literature of the Last 10 Years"

_behavsci, 2023, doi:10.3390/bs13030255_

Round 1

Reviewer 1 Report

The article is interesting and current. There are some concerns about the article. For this reason, I suggest suggestions for improvement. 

In the Introduction, please, explain how the article is structured. Explain in more detail, why it is important to conduct literature review on the topic of telemedicine. 

In chapter 2, please, explain why did you use different combination of Keywords within different databases. 

From the results of the literature review, I can see different approaches of studying telemedicine, but why is this important for the behavior sciences? 

From the discussion and conclusions, I can not see the implications of the study. Why is it important to conduct literature review on telemedicine for behavior sciences? What are limitations of the study? 

Author Response

Dear reviewer, thank you for your comments and feedback to correct. We are infinitely grateful for all suggestions for improvement.

1.In the Introduction, please explain how the article is structured. Please explain in more detail why it is important to conduct a literature review on the topic of telemedicine.

The importance of carrying out a systematic review of telemedicine is currently transcendental because in the world, with SARS CoV2, it was implemented abruptly in all health systems in the world and little by little it was improved and implemented to attend to the health emergency presented. In this sense, the COVID-19 pandemic forced the promotion of the use of telemedicine in different areas of health, which, thanks to current technology, provided new services, innovating and complementing emergency and ambulatory care, which the world is going through (Chá 2020). Moreover, telemedicine is here to stay and therefore must be implemented from many approaches in order to establish protocols that make it a global tool for all citizens of the world in whatever remote region they are in. Therefore, it requires attention from different approaches in order to fill existing gaps such as ethical and legal ones, which will guarantee the implications of a telemedicine service in the world.

The document is structured as follows. Section 2 presents the methodological criteria; section 3 covers both descriptive and narrative results, showing relevant graphs of the systematised articles and their views linked to the research questions. Finally, section 4 develops the discussion and conclusions, presenting a critical analysis of the findings.

Reference

Chá Ghiglia, M. M. Telemedicina: su rol en las organizaciones de salud. Revista Médica del Uruguay, 2020, 36(4), 185-203.

2.In chapter 2, explain why you used different combinations of keywords in different databases.

Different search strategies relevant to the research question and objective were used, and their different approaches and perspectives were discovered with the same syntax of different combinations in the different databases.

3.From the results of the literature review, I can see different approaches to studying telemedicine, but why is this important for the behavioural sciences?

Currently, the use of information and communication technologies is a major challenge in health systems worldwide. With the Coronavirus pandemic disease, regions around the world have implemented Telemedicine services, with varying success, under-lining the need for infrastructure, investment, and regulation in this regard.

It is about providing health services, where distance is an important component, by any health professional, using new communication technologies for the valid exchange of information in diagnosis, procedure and prevention of pathologies or injuries, research and evaluation, and continuing education of health providers, all in the interest of im-proving the health of their individuals and communities.

In terms of human factors, the main challenge facing telemedicine is overcoming resistance to change, which is often multifactorial in nature, such as lack of training in the use of information technologies, fear of the unknown, ethical and legal issues. It is also a reorganization of health systems due to the fact that most health professionals are used to providing face-to-face care, where there is not much experience in distance care, due to lack of education and training in most cases. Now it is also about the workload that health professionals struggle with and their particular interests and beliefs, i.e. what they advocate and put into practice, versus those who do not believe in it. Moreover, these new tasks require training.

Their proper use requires a plan that involves the entire health system, as well as public policies, adequate legislation, and health institutions with the appropriate infra-structure to ensure connectivity and relevant communication sets. All of this involves time and financial resources in addition to the above components.

4.From the discussion and conclusions, I cannot see the implications of the study. Why is it important to conduct a literature review on telemedicine for the behavioural sciences? What are the limitations of the study?

This study is a theoretical contribution to clarify which implications of telemedicine should be considered in order to create a frame of reference to understand what it means to manage its approaches and perspectives where there are areas with a remarkable development in its use, but in others it is still slow and fragmented. It is hoped that in this modality its practical application will be particularly enhanced. This request arises from the need to create a common framework that can be adapted to different settings and pro-vides clear guidance on how to understand telemedicine management. At the same time, it provides analytical perspectives related to the multiple approaches and viewpoints that frame it, highlighting the overall impact of Coronavirus on the process of use and acclimatization to telemedicine, the components that make it feasible and the barriers that impede it.

It is worth mentioning the main limitation of this file. We start with the fact that we are talking about a systematic review that, despite searching as many parts of the literature as possible, could not be fully entered into several databases such as: Wiley. In addition, by not including conference papers, field reports, organizational reports, etc., we have not been able to include any of the literature. Although we strive to minimize author subjectivity, it is possible to assume that important references have been ignored for research purposes when selecting journals with high scientific impact. Despite its limitations, it is hoped that this research will provide important insights for the formulation of further experimental and empirical research.

Reviewer 2 Report

Dear Authors, 

This article is generally well-designed. It will contribute to the literature. I have some suggestions to make it a more targeted and excellent article; 

1- The language of the article needs to be improved (there are grammatical errors and typos). It is recommended that the article is edited by a native English speaker. 

2- The importance of telemedicine in the covid-19 pandemic should also be mentioned. For example, you can read and benefit from the articles below; 

- Temiz, S. A., Dursun, R., Daye, M., & Ataseven, A. Evaluation of dermatology consultations in the era of COVID‐19. Dermatologic therapy, 2020;33(5), e13642.

- Bashshur, R., Doarn, C. R., Frenk, J. M., Kvedar, J. C., & Woolliscroft, J. O. (2020). Telemedicine and the COVID-19 pandemic, lessons for the future. Telemedicine and e-Health, 26(5), 571-573.

Best wishes ...

Author Response

Dear reviewer, thank you for your comments and feedback to correct. We are infinitely grateful for all suggestions for improvement.

1.The language of the article needs to be improved (there are grammatical errors and typos). It is recommended that the article be edited by a native English speaker.

The translation of the paper has been done by a native speaker, however, proofreading, grammatical and typo corrections have been made.

2.The importance of telemedicine in the covid-19 pandemic should be mentioned. For example, the following articles can be read and used;

Bashshur, R., Doarn, C. R., Frenk, J. M., Kvedar, J. C., & Woolliscroft, J. O. Telemedicine and the COVID-19 pandemic, lessons for the future. Telemedicine and e-Health, 2020, 26(5), 571-573. https://doi.org/10.1111/dth.13642

Temiz, S. A., Dursun, R., Daye, M., & Ataseven, A. Evaluation of dermatology consultations in the era of COVID19. Dermatologic therapy, 2020, 33(5), e13642. https://doi.org/10.1089/tmj.2020.29040.rb.

These articles have been included in the theoretical drafting of the document.

The adoption of telemedicine in all health systems around the world, following the global SARS CoV2 crisis, demonstrates its valuable utility as an effective tool to safeguard the so-called social distancing in clinical settings. However, this situation is evidence of the slow adoption of telemedicine throughout history, despite the varied research that has been carried out, as a result of the experience acquired in its application and adoption in different healthcare scenarios, ranging from monitoring to teaching. For this reason, the usefulness of telemedicine should be valued, and it should not only be limited to the management of the current health crisis, but should also be transcended because it has been established abruptly, adopted to the crisis and is here to stay. In this sense, a large number of outpatient consultations in various settings can be managed clinically and efficiently through telemedicine. It also requires the necessary infrastructure with the use of technology, the necessary logistics in terms of material, human and financial resources from the point of view of governments for public health centres and also private investment.

Today, due to the rise of telemedicine, the main actors who have worked in this field cannot rest on their laurels, it is necessary to develop a know-how, with all the events they have experienced and which allowed them the merit of having a vision that enabled them to extend the reach of healthcare resources to those who need them, regardless of distance and time. It is time to maintain the route to safe and effective medical care. More specifically, using a telemedicine outreach to all, where all actors and especially physicians, who must respect standards, institutional protocols, quality assurance mechanisms in place, prompt reporting of adverse events, proper documentation and follow-up through virtual health records (Bashshur et. al., 2020).

During the height of the global health crisis, it was reported that some consultations could be a vector for SARS-CoV2 transmission. Therefore, face-to-face interventions had to be postponed during the pandemic period. In this context, the importance of the adoption, implementation and application of telemedicine in consultations arises. This know-how can help all health systems to ensure telemedicine with global reach and increasingly committed to meeting the needs of populations in all settings around the world (Temiz et al., 2020).

Reviewer 3 Report

This systematic review aims to provide an overview of the approaches and perspectives of telemedicine worldwide. The study focuses on analyzing the theoretical and empirical studies on telemedicine management in the last 10 years, using a comprehensive literature search of 50 articles from multiple databases such as Scopus, Scielo, Ebsco, ProQuest, Dialnet, and Redalyc. The selection criteria for articles included a focus on the last 10 years, being scientific articles, language, variables, and open access. The articles were excluded if they were repeated, did not address the variable, or were not open access.

The results of the study indicate a trend towards managing telemedicine through various approaches and scenarios. These approaches can be grouped into five categories: humanistic, socio-economic, ethical, contingency in the Armed Forces - NASA, and application in the field of medicine with teaching to the entire chain of users. Additionally, the study highlights the importance of controls and monitoring of patients in telemedicine management.

The conclusion of the study is that telemedicine management worldwide is facing challenges that must be addressed in order to overcome the existing barriers and improve access to healthcare systems globally. It highlights the need for a comprehensive and multidisciplinary approach to telemedicine management, taking into consideration the humanistic, socio-economic, ethical, and contingency aspects of telemedicine.

Thank you for taking the time to prepare this article. Below in the form of bullet points are my questions, concerns or things to improve. Please address each of the points in your response to this review and correct any errors identified.

1. Why did you use the less accurate article evaluation system according to Moreno et al (2018)? Instead of using the widespread, proven review system http://www.prisma-statement.org/ ? Could you please add as a supplement to this article only the checklist from PRISMA-STATEMENT. This will check the credibility of your work.

2. Figure 1. there is an error in the caption of the graphic in one of the cells? 4=24? Please check and internally correct.

3. Throughout the metophology, the process of thorough article selection is not described. Were the 18,000 articles reviewed by hand? Have they been reflated according to the criteria indicated in the methodology (not all databases have all the options you indicated). How, then, did you qualify 50 out of approximately 18,000 articles? What software did you use? A sorting system? Filters in the databases? You do not mention anything about methodology. Please correct this, otherwise the article cannot be accepted.

4. Figures 3, 4, 5 contribute nothing to the article. The origin of the words contained in them is not even described in the legends. Are these article keywords or a cluster of words on a topic that makes no sense?

5. The discussion lacks references to similar systematic reviews on the same topic. 

Author Response

Revisor 3

Dear reviewer, thank you for your comments and feedback to correct. We are infinitely grateful for all suggestions for improvement.

1.Why did you use the less accurate article evaluation system according to Moreno et al (2018)? Instead of using the widespread and proven review system http://www.prisma-statement.org/ ? 2. Could you add as a supplement to this article only the PRISMA-STATEMENT checklist. This will check the credibility of your work. It has worked with PRISMA 2020, and this will be reflected in the new version delivered.

Systematic reviews are important in many critical and useful aspects of the evolution of the world's systems. They provide a synthesis of the state of knowledge in a given area, from which important and future research priorities can be identified, address questions that cannot otherwise be answered by individual studies, identify problems in primary research that need to be corrected in future studies, and generate or evaluate theories about how or why phenomena of interest occur. In that sense, the PRISMA 2020 statement (used in this research) has been designed primarily for systematic reviews of studies evaluating the effects of health interventions, regardless of the design of the included studies (Page et al., 2021). In that sense, the present systematic review is structured taking into account the main items of the PRISMA 2020 guideline.

Page, M. J., McKenzie, J. E., Bossuyt, P. M., Boutron, I., Hoffmann, T. C., Mulrow, C. D., ... & Alonso-Fernández, S. Declaración PRISMA 2020: una guía actualizada para la publicación de revisiones sistemáticas. Revista Española de Cardiología, 2021, 74(9), 790-799.

  1. Figure 1. is there an error in the chart legend in one of the cells? 4=24? Please check and correct it internally.

The error in the figure was checked and corrected. Thank you for your comment.

  1. The entire methodology does not describe the process of careful selection of the articles. Were the 18,000 articles reviewed by hand? Were they reflowed according to the criteria indicated in the methodology (not all databases have all the options you indicate)? How did you then rate 50 of the approximately 18,000 articles? What software did you use? A ranking system? Filters in the databases? You do not mention anything about methodology. Please correct this, otherwise the article cannot be accepted.

A detailed explanation was added to the methodology.

The review began with a search in each of the databases, with their respective keywords and search strings or strategies of interest for the large variable and/or category of study. Initially, a large amount of research was found, so a first selection was made where the search strategies were expressly contained in the title, leaving a smaller number, and then the inclusion and exclusion criteria were established, such as the years of publication, the content of the categories to be analysed and the availability of each article (open access). Throughout this process, the Microsoft Excel tool was used to organise all the downloads allowed by the databases, their respective summaries and with the help of text formulas and internal searches, the articles were pre-selected for reading and downloading according to DOI or URL link, respecting the criteria and objectives of the research, also excluding duplicate articles. A more exhaustive review was then carried out, i.e. the articles were selected by approach and, within each approach, the content of each article was prioritised. Finally, 50 articles addressing telemedicine from 5 approaches in the last 10 years were systematised. These databases were Scopus, Ebsco, Scielo, ProQuest, Redalyc, and Dialnet; between the years 2012 - 2022.

  1. Figures 3, 4, 5 contribute nothing to the article. The origin of the words they contain is not even described in the captions. Are they key words of the article or a bunch of words on a meaningless topic?

The objective of each word cloud reflected in figures 3, 4 and 5 was to show the most salient "term" or "key word" of each approach, processed from the summaries elaborated by the authors, with respect to the systematised articles, but linked to each approach, in order to verify which were the predominant ones.

However, we accept your comment and will proceed to eliminate each of these figures in order to focus on the narrative explanation of each approach only.

  1. The discussion lacks references to similar systematic reviews on the same topic.

We have proceeded to review the systematised articles used in the discussion and added some other points of analysis, as well as other articles from systematic reviews to respond to your kind comment and enrich the discussion. The PRISMA 2020 methodology allows other documents to be added to strengthen the argumentation.

Humanistic approach

Furthermore, this approach is supported by a study by Nieblas et al. (2022) who argue that pandemics pose significant challenges to health care, especially in vulnerable countries such as those in Latin America, which experienced during SARS CoV2 a high occupational risk generated by the saturation of health services and the shortage of personal protective equipment (PPE) for health care workers. This required the implementation of strategies to respond efficiently to the health emergency. Compared to developed countries, which had more experience, telemedicine was practised in parallel to conventional care long before the pandemic. However, in developing countries, the absence of technological resources and lack of platforms for telemedicine consultations hindered its use and development.

Nieblas, B., Okoye, K., Carrión, B., Mehta, N., & Mehta, S. Impact and future of telemedicine amidst the COVID-19 pandemic: a systematic review of the state-of-the-art in Latin America. Ciência & Saúde Coletiva, 2022, 27, 3013-3030.

Socio-economic approach

On the other hand, we have the socio-economic approach, which is supported by Albornoz-Chauca et al. (2022) who state that the impact of the COVID-19 pandemic aggravates the vulnerability of rural areas. Access to health care is an important variable in health for the health organization that must provide timely emergency services, health care is an urgent and socially relevant issue in all countries. Therefore, the use of telemedicine in rural health care in times of pandemics for disease control is one of the major issues facing rural communities due to the scarcity of resources, the long distances over which they are geographically located, and the deterioration of health care facilities evidenced in recent years. In this sense, telemedicine is called upon to solve this problem and, at the same time, to help reduce the health gaps that exist between rural and urban areas, which is the great disparity between access to health care and general health of rural inhabitants compared to urban dwellers.

Albornoz-Chauca, M., Gamboa-Cruzado, J., Montero, J. N., Pérez-Salcedo, R., Plata, C. G. -., Yauris-Silvera, C., . . . Elias-Silupu, J. Telemedicine and its impact on rural health care in times of COVID-19: A systematic review. Boletin De Malariologia y Salud Ambiental, 2022, 62(2), 171-182. doi:10.52808/BMSA.7E6.622.007

Medical Approach: Experiences of Interventions around the World

On the other hand, the medical approach is the one that has the most development in the world through different telemedicine interventions globally, thus De Correia et al. (2021) in a study sought to determine the effectiveness of telemedicine in the provision of diabetes healthcare services in low and lower middle-income countries. They conducted a selective literature search, extracting data on study characteristics, key endpoints, and outcome effect sizes. Then, using random effects analysis, they performed a series of meta-analyses for both biochemical outcomes and patient-related properties. They concluded that: Although telemedicine was found to be effective in improving several diabetes-related outcomes, the certainty of the evidence was very low due to considerable heterogeneity and risk of bias.

Correia, J. C., Meraj, H., Teoh, S. H., Waqas, A., Ahmad, M., Lapão, L. V., . . . Golay, A.  Telemedicine to deliver diabetes care in low-and middle-income countries: A systematic review and meta-analysis. Bulletin of the World Health Organization, 2021, 99(3), 209-219B. doi:10.2471/BLT.19.250068

Contingency approach

On the other hand, Mohammadi et al. show the application of telemedicine in various remote locations by the armed forces. In other words, military medicine is an academic discipline with broad applications and fields. Because military physicians often practice a specific body of knowledge of the medical problems and needs of the armed forces, which is often different from general medicine. Furthermore, in their results they found that videoconferencing and e-mail were the main means of communication in telemedicine. Therefore, military medicine, given the shortage of specialists and human resources, needs technology and should be equipped with videoconferencing equipment to improve the quality of health services. Also, the lack of studies related to telemedicine in military medicine was one of the main limitations of this research (Mohammadi et al, 2020).

Mohammadi, R., Tabanejad, Z., Abhari, S., Honarvar, B., Lazem, M., Maleki, M., & Garavand, A. A systematic review of the use of telemedicine in the military forces worldwide. Shiraz E Medical Journal, 2020, 21(11), 1-8. https://brieflands.com/articles/semj-99343.html

Round 2

Reviewer 1 Report

The paper can be accepted.

Author Response

Dear Reviewer 
We sincerely appreciate your initial comments and observations, as well as your final approval of our document.

Best wishes for your health and well-being.

Yours sincerely.

Reviewer 3 Report

Thank you for your reply. 

1. The authors did not include the PRISMA report in the non-significant paper (mentioned in the previous review in points 1 and 2), which supports the idea that the systematic review may not have taken place and that the data that are presented in the non-significant paper may be fabricated.

As other systematic review reports are not included in this paper, it cannot be considered a systematic review. In addition, this article has not been registered in any databases that collect information on systematic reviews.

2. The authors have corrected all image comments.

3. In their response, the authors write: "Initially, a large number of studies were found, so a first selection was made where search strategies were explicitly included in the title, leaving a smaller number, and then inclusion and exclusion criteria were established (...)." Which raises a big suspicion. Because when you type in the phrases the authors mention (I personally checked in several databases) in the medical databases the authors cite, the result is hundreds of thousands of results. It is not physically possible, without appropriate software, to go through all the titles, let alone familiarise oneself with the abstracts The authors clearly indicate in their response that it was only after checking the titles of the articles that the inclusion and exclusion criteria were introduced.

The above response again begs the question, was the review done in a systematic way?

How many people carried out this review? At what time was it done?

4. The authors have made extensive changes to the discussion, which add a lot to the article.

Nonetheless, I believe that the article bears no signs of a systematic review due to all the comments made in this and the previous review and the lack of registration of the review in the relevant databases, and the lack of reporting (there is no supplement attached to the article to show that the review was done in a systematic way. Therefore, the title should be replaced and the article itself should be categorized as a narrative review or literature review.

Author Response

Revisor 3

Segunda ronda

  1. The authors did not include the PRISMA report in the non-significant paper (mentioned in the previous review in points 1 and 2), which supports the idea that the systematic review may not have taken place and that the data that are presented in the non-significant paper may be fabricated. As other systematic review reports are not included in this paper, it cannot be considered a systematic review. In addition, this article has not been registered in any databases that collect information on systematic reviews.
  2. In their response, the authors write: "Initially, a large number of studies were found, so a first selection was made where search strategies were explicitly included in the title, leaving a smaller number, and then inclusion and exclusion criteria were established (...)." Which raises a big suspicion. Because when you type in the phrases the authors mention (I personally checked in several databases) in the medical databases the authors cite, the result is hundreds of thousands of results. It is not physically possible, without appropriate software, to go through all the titles, let alone familiarise oneself with the abstracts The authors clearly indicate in their response that it was only after checking the titles of the articles that the inclusion and exclusion criteria were introduced. The above response again begs the question, was the review done in a systematic way? How many people carried out this review? At what time was it done?

We thank you for your comments and then go on to respond to points 1 and 3.

If the review has been carried out, with the terms indicated, it would not be possible to speak of a fabrication or invention of a process, as we are extremely respectful of ethical principles. We also consider the article to be significant for all the reasons given in each section and in the conclusions reached, considering that it has already been approved for publication by the other reviewers.

We understand that the wording of the process and the grouping initially reported (and which you have logically tried to replicate) of the search terms suggest that the search strategy is a total combination of what is reported therein, however, they were made at various times and with the filters and criteria already expressed in the methodology and are in number those reported in figure 1.

However, we have adjusted the wording and accepted your recommendation to better fit a literature review, which was conducted by the authors, over a period of four months between August and November 2022.

The wording in the document in the methodological part would read as follows:

The review began with a search in each of the databases, with their respective keywords and search strings or strategies of interest for the large variable and/or category of study.  Initially, a large amount of research was found, so a first selection was made where the search strategies were expressly contained in the title, leaving a smaller number, together with several inclusion and exclusion criteria were established, such as the years of publication, the content of the categories to be analyzed, and the availability of each article (open access), among others.

Throughout this process, the Microsoft Excel tool was used to organize all the downloads allowed by the databases, their respective abstracts and with the help of text formulas and internal searches, the articles were pre-selected for reading and downloading according to DOI or URL link, respecting the criteria and objectives of the research, also excluding duplicate articles. This was followed by a more exhaustive review, i.e. articles were selected by approach and, within each approach, the content of each article was prioritized. Finally, we worked with 50 articles addressing telemedicine from 5 approaches over the last 10 years. These databases were Scopus, Ebsco, Scielo, ProQuest, Redalyc and Dialnet; between the years 2012 - 2022.

Below, we show the search terms used (as a grouped report of everything searched), highlighting their relevance to the research question and objective, and discovering their different approaches and perspectives in the different databases: Telemedicine, telemedicine management, telehealth, eHealth, implematatation, telemedicine system.

  1. The authors have made extensive changes to the discussion, which add a lot to the article.

Nonetheless, I believe that the article bears no signs of a systematic review due to all the comments made in this and the previous review and the lack of registration of the review in the relevant databases, and the lack of reporting (there is no supplement attached to the article to show that the review was done in a systematic way. Therefore, the title should be replaced and the article itself should be categorized as a narrative review or literature review.

We understand and respect very much your comments and suggestions, we have corrected the wording of the methodology to better understand what was done. We understand that the wording of the process and the initially reported grouping (and that you have logically tried to replicate) of the search terms suggest that the search strategy is a total combination of what was reported there, however, they were made at various times and with the filters and criteria already expressed in the methodology and are in number those reported in figure 1. An additional issue is how the document was translated, as it was originally done in Spanish, but apparently when translated there was a distortion of the process that led to a misunderstanding, which we have already reviewed and corrected for approval.

We accept the suggestion to change it to a literature review, in order to report the important findings. We consider it wise and prudent and will take it into consideration for future revisions.

Thank you very much for your kind comments.

Round 3

Reviewer 3 Report

I thank the Authors for all their responses.

This article has now gone from being a systematic review to a review, and a selective one at that.

After any changes made by the authors, of which there were quite a few, this paper can be considered a good one. Nevertheless, it does not contribute much to the field covered in the article. It does, however, provide a selective summary of some important or not aspects.